# Philosophy of a “Good Death” in Small Animals and Consequences for Euthanasia in Animal Law and Veterinary Practice

**DOI:** 10.3390/ani10010124

**Published:** 2020-01-13

**Authors:** Kirsten Persson, Felicitas Selter, Gerald Neitzke, Peter Kunzmann

**Affiliations:** 1Stiftung Tierärztliche Hochschule Hannover, Bünteweg 9, 30559 Hannover, Germany; Peter.Kunzmann@tiho-hannover.de; 2Medizinische Hochschule Hannover, Carl-Neuberg-Straße 1, 30625 Hannover, Germany; Selter.Felicitas@mh-hannover.de (F.S.); Neitzke.Gerald@mh-hannover.de (G.N.)

**Keywords:** veterinary ethics, euthanasia guidelines, moral stress, animal death

## Abstract

**Simple Summary:**

Euthanasia in veterinary practice is often discussed as one of the profession’s major burdens. At the same time, it is meant to bring relief to terminally ill and/or severely suffering animal patients. This article examines “euthanasia” from a philosophical perspective regarding different definitions and underlying basic assumptions concerning the meaning of death and welfare for nonhuman animals. These theoretical issues will then be discussed in relation to laws and guidelines on euthanasia and practical challenges with end-of-life decisions in small animal practice. Factors which are identified as potential causes of the complex problems regarding euthanasia are as follows: the confusing framework for euthanasia in law and soft regulations; the inclusion of many stakeholders’ perspectives in end-of-life decision-making; potential conflicts between the veterinarians’ personal morality and legal requirements and professional expectations; and, most of all, the veterinarians’ lack of awareness for underlying philosophical assumptions regarding possible understandings of euthanasia. Different practical suggestions are made to clarify and facilitate euthanasia in small animal practice.

**Abstract:**

Moral stress is a major concern in veterinary practice. Often, it is associated with the challenges in end-of-life situations. Euthanasia, however, is also meant to bring relief to animal patients and their owners. The reasons for the moral strain euthanizing animals causes to professional veterinarians need to be further clarified. This article investigates “euthanasia” from a philosophical, legal, and practical perspective. After introducing relevant aspects of euthanasia in small animal practice, the term is analyzed from an ethical point of view. That includes both a broad and a narrow definition of “euthanasia” and underlying assumptions regarding different accounts of animal death and well-being. Then, legal and soft regulations are discussed with regard to the theoretical aspects and practical challenges, also including questions of personal morality. It is argued that the importance of ethical definitions and assumptions concerning euthanasia and their intertwinement with both law and practical challenges should not be neglected. The conclusion is that veterinarians should clarify the reasons for their potential discomfort and that they should be supported by improved decision-making tools, by implementation of theoretical and practical ethics in veterinary education, and by updated animal welfare legislation.

## 1. Introduction

Euthanasia is an established part of veterinarian practice. The “good death” (Greek “*eu*” and “*thanatos*”) is meant to bring relief to animals and their owners. Nevertheless, many veterinarians consider it to be one of the profession’s major burdens, and many owners of companion animals feel overwhelmed by the need to make a decision on behalf of vulnerable individuals. On the one hand, the phenomenon of moral stress [1] and the alarmingly high suicide rate among veterinarians are often associated with their professional obligation to kill their patients [2,3,4]. This, on the other hand, is not reflected in the concept of euthanasia as a powerful tool that has been at the veterinarians’ (while not at the human medical doctors’: Human euthanasia is illegal in most countries. Exceptions are Benelux, Canada, and Colombia.) disposal for good reasons.

This article investigates the interrelations of philosophical considerations with laws and guidelines, personal moral attitudes, and practical context and constraints as the potential causes for the striking ambivalence that accompanies euthanasia in small animal practice.

The frequency and diversity of euthanasia cases are dependent on both the legal framework of a country and the veterinarian’s working environment. What is perceived as stressful or burdensome, however, can also be based on motivational or other psychological properties of the individual veterinarian. Representing an American perspective on the issue, Rollin [1] claims that the crucial conflict lies in the discrepancy between the reasons for choosing the profession, these being the animals’ advocate, helping animals, etc., and the necessary choices and actions of a veterinarian’s daily routine, like having to decide in favor of the owner’s instead of the patient’s interests, killing surplus animals in shelters, and so forth. This assertion is especially true for those countries in which “convenience euthanasia” is legal and not uncommon.

There are, however, further potentially influential factors that contribute to the field of tension concerning end-of-life (EOL) decisions in small animal practice.

Despite euthanasia being a common process in veterinarians’ professional practice, research suggests that veterinarians do not feel well-prepared for the task during their theoretical education [5,6]. On the one hand, some veterinary students report decreasing empathy with animals during the course of their studies [7,8], which could facilitate coping with euthanasia, as trained veterinarians feel more detached from animals. On the other hand, euthanasia is one of the most frequently mentioned challenging or dilemma situations in small animal practice [9]. It is also judged to be most stressful [10,11], and, occasionally, performing euthanasia is even refused for moral reasons [12]. Although veterinarians are familiar with the legal framework, they often cannot name the relevant criteria or tools that they use in EOL decisions [3,9], and few are convinced that further development of such tools would be necessary and helpful [5]. The discourse on implementing more training regarding communication and ethical judgements skills in veterinary education cannot be fully elaborated here. Nonetheless, there is apparently some request for improvement [13,14,15].

A fundamental challenge, however, is also presented by the tension of the motivation underlying the choice to become a veterinarian in many cases; helping animals and improving their lives, on the one hand, and the mere act of taking their lives, leaving aside reasons and justifications, as part of the job as a veterinarian. It is, therefore, important to differentiate between moral stress due to killing animals per se (which makes euthanasia challenging per se) and moral stress due to killing animals *for the wrong reasons* (which asks for precise criteria for justified euthanasia); a difference that is not always clearly separated in the literature.

Furthermore, it is necessary to distinguish between a veterinarian’s patient-centered ethical approach (i.e., animal-centered or “care orientation” [16] (p 2)) that is led by empathy for the vulnerable individual and a balancing ethical approach (or “justice orientation” (ibid.)) that is based on the weighing of factors of all involved individuals, aiming at a fair outcome. While the veterinarian’s role of the animals’ advocate is still dominant in, for example, German-speaking European countries, the “Unit-of-Care” is an upcoming concept of the American animal hospice movement [17,18]. The latter focusses on the relationship between the companion animal and its owner and not only takes into consideration the animal’s needs, but also, for example, the owner’s religious beliefs or good death ideals.

We thus argue that the problem of moral stress linked to EOL decisions in small animal practice is as follows:Multilayered;Linked to psychological factors like professional motivation;Linked to conceptual confusion regarding uses of “euthanasia”;Linked to a lack of reflection of theoretical differentiations in laws and guidelines.

Having introduced current challenges with EOL decisions in veterinary practice (cf. 1. Introduction, we next examine euthanasia from a philosophical perspective, regarding different definitions, connotations, and basic assumptions (cf. 2. Euthanasia in Ethics). In a third step, laws and guidelines on euthanasia will be considered in relation to the practical and theoretical issues raised in the first two sections (cf.3. Euthanasia in Laws and Guidelines). Finally, the analysis will be discussed, addressing a number of specific problems that emerge for veterinarians in the context of euthanasia.

## 2. Euthanasia in Ethics

Justification for killing nonhuman animals is a crucial issue in animal ethics. Amongst other contexts, such as slaughtering, self-defense, culling, or killing in research and testing, euthanasia is prima facie considered ethically justifiable. However, this judgement depends on different factors: first, there are two fundamentally different definitions of euthanasia; second, these different definitions indicate multiple underlying accounts of the meaning of death and, third, different accounts of welfare for nonhuman animals. These distinctions are also related to different criteria for a death being “good”.

### 2.1. There Are Different Definitions of “Euthanasia”

When used in human medical ethics, the widely accepted meaning of the term “euthanasia” depends on the context, such as a country’s legal guidelines and medical practice. It is a blurry and loaded concept that requires semantic clarification ahead of ethical debate [19]. It may be helpful to state the same regarding the euthanasia discourse in veterinary medicine, as it is used for procedures which differ in ethically relevant aspects. There are even requests for a new term or terms to provide more precise linguistic concepts when talking about killing animals: “The term euthanasia, defined as an act which fulfils the interest of the one who will die and motivated by a moral imperative, applies to one form of morally justifiable killing of animals, but we need terminology that recognises the distinction between these and poorly justified or ethically unjustifiable killing of animals” [20] (p 217).

#### 2.1.1. At Present, There Is a Broader Definition that Explicitly Widens the Literal Meaning 

In the *Encyclopedia of Animal Rights and Animal Welfare*, it reads, “The definition of euthanasia differs slightly in veterinary medicine and human medicine. In human medicine, the term is restricted to ‘mercy killing’—killing a patient when death is a welcome relief from a life that has become too painful or no longer worth living. The definition is broader in veterinary medicine, however, including as well the euthanasia of healthy animals for owner convenience, for reasons of overpopulation, for behavior problems, or as donors of tissues [sic] for research” [21] (p 164).

Clearly, this definition is taken from the context of application and focuses on the action of euthanizing. The reference to veterinary medicine suggests that euthanasia is conducted for reasons which lie far beyond the patient’s best interest, in a strict sense. Thereby, this definition is descriptive and formed empirically rather than being prescriptive and guiding. Despite the concept’s problematic origin, the broad definition is mirrored in some guidelines (cf. 3. Euthanasia in Laws and Guidelines.), pointing out that euthanasia is performed when a veterinarian intentionally kills a patient independent of the patient’s interest. Defined in that sense, the term “euthanasia” is problematic in different ways: First, it lacks the normative impact that is strongly intertwined with the account of providing a “good death” for a patient, unless “good” is understood in a very weak, instrumental sense, meaning “painless” and “following a standard protocol”. If death caused by a veterinarian is by definition a good death—possibly because it is painless—the veterinarian does not gain any decision criteria besides the methodically correct execution of the euthanasia process. The second problem of the definition is therefore the shift in responsibility to the veterinarian. If it is his/her action that transforms the killing of a companion animal into a good death, he/she needs to decide on the basis of further, unspecified criteria if and when euthanasia is (not) indicated. A third aspect is the addressed contrast to euthanasia in human medicine. If humans are euthanized by medical doctors in their best interest (following their individual request), whereas nonhuman animals are euthanized by a veterinarian for any reason, the following issues must be raised: If animals do not have a best interest regarding a prolongation or ending of their lives (ontological difference);If it is not possible to determine the animals’ best interest (epistemic problem); orIf there is a distinct best interest but it does not have to be considered or is outweighed by the owner’s interest in the process of euthanasia (moral difference).

To bring up further confusion regarding the use of the word, there is the account of “involuntary euthanasia” used for crimes against humanity, such as in World War II, but also in medical ethics, when a patient did not consent to euthanasia despite being able to give consent. Additionally, the term “nonvoluntary euthanasia” is used when a patient is not or no longer able to give consent, but death is presumed to be in the patient’s best interest. If the terms can correctly be applied to nonhuman animals, euthanasia performed in veterinary practice should be classified as either nonvoluntary (when relieving animals from suffering) [22] or involuntary (when animals are killed for other reasons). In contrast to that, Yeates [23] suggests that the relevant criterion here lies in the owner’s will, in a way that “involuntary euthanasia” refers to situations where there is no agreement or disagreement to euthanasia by a (potential) owner, “nonvoluntary euthanasia” should be used to indicate that the animal is euthanized against the owner’s wishes and “voluntary euthanasia” for those instances where the owner agreed to euthanize the animal.

It is, however, more plausible to fully abandon those terms when talking about nonhuman animals, as they refer to the absence of autonomous human consent. Animals are nonautonomous patients and thereby per se unable to give consent—or to use Cholbi’s words [22] (p 266) “[a]nimals do not consent to their own death, nor would it make sense to ask them to do so”. If they cannot be euthanized voluntarily, the use of the terms “involuntary” or “nonvoluntary” is questionable. Not having access to their potential interest in dying or staying alive makes using the term “involuntary” euthanasia—at best—tautological. Furthermore, it makes euthanasia of companion animals a very different act from the euthanasia of humans: We assume that a per se nonautonomous creature would consent to be euthanized if it had all the information and capacities we have, knowing, at the same time, that it is exactly the difference between us (humans) and it (the nonautonomous animal) that makes a huge difference in the decision.

#### 2.1.2. There Is a Narrow Definition in the Literal Sense of the Term

If euthanasia is understood in the literal sense as “proper euthanasia” [24] (p 21), it is defined as the (painless) killing of an individual by a veterinarian, if ending its life is presumably in the individual’s interest. The focus here is rather on the intention when euthanizing than on the action itself.

This narrow and less counter-intuitive definition is, likewise, demanding regarding several aspects. However, it is based on the following assumptions: Firstly, animals do have an interest in not continuing their lives under certain circumstances (ontological premise);Or animals do generally not have an interest in prolonging their lives but only in minimizing their negative mental states and maximizing their positive mental states or following their “telos” [25] (p 49) (ontological premise);Secondly, veterinarians and/or owners have access to the individual animal’s interests and are therefore able to weigh the companion animal’s suffering against its interest to live on (epistemological premise);Or humans do not have access to the animal’s distinct interests, but they are sufficiently competent to approximately find the point of time when continuing to live—objectively—would be worse than to die and are therefore permitted and in specific cases even obligated to end an animal’s life (life comparative account according to Cholbi [22]).

The broader as well as the narrow definition need to be accompanied by ethical considerations. The broader definition points towards the veterinarian’s ethical competencies, as he/she is the one to professionally judge cases and conduct the procedure correctly according to prescribed/formal standards. Economic and practical constraints are crucial aspects of the decision-making process, especially when considering laboratory and shelter animals.

These factors lead to a potential third definition suggested by Yeates [23], which he calls “‘contextually-justified euthanasia’ where an animal could have a life worth living in an ideal world, but the circumstances mean that that opportunity is not worthwhile. This may be due to an owner’s unreasonableness or the fault of society, but the veterinarian should not feel guilty for ‘making the best of a bad job’” (p 71). Yeates’s way of putting emphasis on the context of the case is strongly reflected in guidelines for end-of-life decision in companion animals (cf. 3. Euthanasia in Laws and Guidelines.). As much as this effort can be practically and psychologically helpful for veterinarians, it might prevent thorough reflection on factors that are crucial for euthanasia in the narrow sense, e.g., the animal’s presumed best interest and quality of life.

The debate regarding the narrow definition of euthanasia (“in their own interest”) touches very basic questions of animal philosophy and cognition (especially their account of their future life) and animal ethics (weighing up the harm of suffering and the harm of death). Once the abovementioned underlying assumptions are accepted (ontological premise and epistemological premise or life comparative account), the EOL debate focusses mainly on the best point of time to end an animal’s life rather than on the question if euthanasia is in the animal’s interest at all.

### 2.2. There Are Different Accounts of the Meaning of Death for Nonhuman Animals

For the narrow definition of euthanasia, it is important to clarify the meaning of death for nonhuman animals. Although it is questionable—like for many other issues in animal ethics— whether general claims can be made for all kinds of animals, the patients in small animal practice represent a rather homogenous group (mainly cats, dogs, other small mammals, and birds) regarding their potential perception of death and dying.

There is no scientific and conceptual agreement on the meaning of death—and, consequently, of killing—for nonhuman animals. Judgements whether death harms an animal are dependent on the account of harm, which is a thick concept (see, for example, different meanings of harm in Belshaw [26]). The discourse spans quite a spectrum of views, which is why only exemplary positions are presented here. For a more comprising discussion, see, for example, Harman [27].

#### 2.2.1. Death Does Not Matter Morally

At one end of the spectrum, there is the claim that (individual) animal death does not matter morally. A prominent utilitarian view is known as the replaceability argument, as Singer explains it for fishes [28] (p 126). Death of a sentient being that lacks certain cognitive capacities does not mean harm as long as the living being is replaced by another being of the same kind. Welfare in this perspective is not linked to a specific individual but only taken into account overall. As long as the next individual reaches roughly the same amount of overall well-being, therefore, it is not morally relevant if the former individual is killed. Death, in that account, does not harm a merely sentient animal. For criticism of this view, see, for example, Višak [29].

#### 2.2.2. Death Is the Greatest Harm

In contrast to that, it can be suggested that death is the most severe harm to animals, even if they have no understanding of death themselves. According to this position, they are living organisms that strive to go on living [30] or that have an inherent worth [31]. This view is detached from the subjective view of the animal individual. The argument why living organisms as such should be direct moral objects or have an inherent worth can only be found in the general attitude of a biocentric worldview which is metaphysically demanding. Schweitzer’s approach is critically discussed in more detail, for example, by Eck [32].

#### 2.2.3. Death and Suffering Must Be Weighed

There are several moral theories on the killing of animals that take a middle ground between these two extreme positions. Two of these are the life comparative account and the time-relative interest account. The former suggests that death is prima facie harmful to animals and that the harm of death has to be weighed against the harm of suffering accompanying the longer life (e.g., Cholbi [22]). Death can be harmful or beneficial depending on the prospective quality of life. On the one hand, death deprives the individual of potentially good experiences; on the other hand, it can prevent an individual from living a life full of suffering. Even if it is not clear whether nonhuman animals might be able to develop a desire to die under certain circumstances, death can be in their best interest. There is, additionally, an ongoing debate, if (and if yes, which) nonhuman animals are able to make plans or have expectations regarding their future lives. These abilities, in turn, become morally relevant in accounts such as the so-called time-relative interest account [33], which considers not only the mere loss of future positive mental states but also the individual’s degree of psychological connectedness between their current and their future self. For a more elaborate discussion of this debate, see, for example, Selter [34].

#### 2.2.4. Death Harms the Animal, but that Is Not Morally Relevant

Alternatively, Belshaw [26] suggests it is possible to acknowledge that death harms the animal, while also questioning the moral relevance of this observation. As animals are unable to have categorical desires, their death has the same moral relevance as the death of plants: It is harmful to the individual, which can be morally significant on an indirect or instrumental level, but not in a sense of direct moral relevance regarding the dying individual. Using harm in this purely descriptive sense is a clear outlier in the face of the abovementioned positions that use “harm” as a normative term, implicitly stating that it is something a moral agent generally wants to prevent.

### 2.3. There Are Different Accounts on the Meaning of Welfare/Well-Being for Nonhuman Animals

In EOL situations, the patient’s (presumed) quality of life is one of the crucial criteria for decisions. Similar to medically incapacitated humans, there is no way of directly accessing an animal’s quality of life. Potentially relevant parameters are subject to scientific and (critical) philosophic investigations in animal-welfare studies [35,36,37]. However, there is an ongoing debate on the best way and the best person to judge an animal’s subjective well-being. On the one hand, the patient’s owners, although being those who know a companion animal best, might be biased by their anthropomorphic conception of the animal’s preferences, emotions and well-being [38]. In addition to that, their emotional links with the animal might cloud their judgement. For veterinarians, on the other hand, the concept of animal welfare is usually linked to farm animal welfare and the five freedoms [39]. Still, they are attributed Aesculapian authority, which includes the final saying when it comes to dilemma situations. Carrying that responsibility, they should be aware that there is a range of accounts on animal welfare (objective), animal well-being (subjective), and animal quality of life.

#### 2.3.1. Narrow Hedonism

In narrow hedonism, the focus is on the avoidance of negative welfare states such as pain, stress, suffering and lack of fulfilment of basic needs. Although narrow hedonism is implemented in prominent animal welfare concepts such as the five freedoms, it is criticized for its logical consequences: “If avoiding suffering was truly all that mattered, then every animal should be killed as soon as possible, since this would ensure the absence of suffering” [24] (p 20). See also Fawcett [20].

#### 2.3.2. Broad Hedonism

Therefore, the concept of broad hedonism suggests considering both negative and positive mental states when evaluating an animal’s welfare. If we take positive mental states into account, then we might question cases in which a painless death is considered permissible (but not obligatory), for example when killing surplus animals in shelters. Ending a life with net positive well-being might be seen as doing harm when adopting an account of broad hedonism [24] (p 23). This view is also supported by the suggestion of promoting a “live worth living” [40] (p 32) in addition to preventing a “live worth avoiding” (ibid.).

#### 2.3.3. Quantity Versus Quality of Life

Much like in EOL debates in human medicine, there is a challenging weighing process, linked to the assumption that keeping a patient alive should always be prioritized: prolonging life without increasing quality of life might cause more harm than good. With progress in veterinary medicine and the attitude to consider companion animals as family members, it is possible to keep animals alive until a very old age or until a disease has progressed quite far. If being alive is not considered a value in itself (cf. 2.2.2. Death Is the Greatest Harm), the quality of life has to be measured separately [1].

## 3. Euthanasia in Laws and Guidelines

Dealing with veterinarians’ practice, laws and guidelines that establish rules for euthanasia introduce even further perspectives, i.e., the animal owner’s and also public interests. The guidelines implicitly answer the question to which degree the justification of euthanasia may be based not only on the perceived animal’s interests but also on human interests, especially on those of the animal’s owners, as well as public financial or safety interests. Legislation and soft regulation in this respect offer an interesting range of verdicts: regulations, e.g., in Germany and Austria consider the protection of animals’ lives as an objective of the law and therefore demand a “good reason” [41] to end this life. In other cultures, EOL decisions in the case of animals are based more clearly on human interests.

The general shift in societal human–animal relations presents a challenge to existing laws and it has not yet been sufficiently implemented. Correspondingly, upcoming requests for justifications of morally relevant actions involving animals present a contrast to the outdated, however legally manifested, assumption that it is self-evident that animals are at humans’ command.

In companion animals, the concept of “convenience euthanasia”, i.e., euthanizing a pet against her or his presumed interest but merely due to the patient’s owner’s wishes, is legally prohibited in German-speaking countries, but common routine elsewhere (cf. 1. Introduction). However, there are cases in which the given context questions a clear distinction between cases of euthanasia in the animal’s presumed best interest and those of euthanasia in the owner’s but not the animal’s interest, for example, as subsumed in Yeats’s [23] account of “contextually justified euthanasia”.

Euthanasia is regulated in the Animal Welfare Acts of several countries worldwide (considered in this article: German, Swiss, Austrian, Swedish Animal Welfare Law). Additionally, there are legally nonbinding regulations by nongovernmental associations and organizations (considered in this article: Guidelines by the Australian Veterinary Association, Ethikkodex TÄ Deutschland, Positionspapier der Schweizerischen Vereinigung für Kleintiermedizin, Ethische Grundsätze für den Tierarzt und die Tierärztin GSTSVS, British RCVS, AVMA Guidelines on Euthanasia, Decision tools by BVA’s Ethics and Welfare Group, Entscheidungshilfe Euthanasie bei Kleintieren [42]). Depending on their definition of euthanasia, they provide criteria like diagnosis, prognosis, nonmedical factors concerning the animal, the owner’s convenience, financial constraints, and potential further interests, to guide the decision-making process. While all laws and guidelines agree on some form of euthanasia being ethically justifiable, there is a broad spectrum of legally accepted good reasons. Most of those reasons are derived from a sentientist perspective, giving highest priority to the avoidance of animal suffering (narrow hedonism, cf. 2.3.1. Narrow Hedonism) and either denying the harm of death for nonhuman animals or weighing it as less important (cf. 2.2 Different Accounts of the Meaning of Death). The wording generally suggests a narrow account of euthanasia (AVMA Guidelines for the Euthanasia of Animals 2013; RCVS 2019).

An alternative understanding of the term is provided by the former EU Recommendations for Euthanasia of Experimental Animals. from 1997 [43] (p 3) , suggesting that euthanasia is performed if it is “an act of humane killing with the minimum of pain, fear and distress” without reference to the patient’s interest, but adding the objective that death should be “appropriate for the age” (ibid.). The current EU directive on this matter completely abandons the term “euthanasia” and provides specifics on “humane end-points” [44] (recital 14) and methods of killing, instead. The killing of laboratory animals can be called euthanasia if the broad definition of euthanasia is adopted. Generally, the term “humane killing” is more adequate when referring to killing an animal without pain and distress in a context of animal research and testing. Whether killing can also be in the laboratory animals’ presumed best interest (narrow definition) cannot be discussed in full here.

In other regulations, the inclusion of paragraphs like “destruction of ‘dangerous’ dogs” [45] (p 57) or expressions like “dealing[…]with the euthanasia of healthy, unwanted animals[…]” [46] (p 15) suggests that, at least implicitly, most regulations are based on the abovementioned broad definition of euthanasia, despite their rather explicit references to the animals’ best interests.

Although decision trees and algorithms are neither appreciated nor frequently used or asked for by the majority of veterinarians [47], there are numerous suggestions how to sort factors and criteria that are and should be considered in EOL decisions. These are often directly connected to the legal scope of a country. The British Veterinary Association, for example, provides an algorithm including questions like “Is the benefit to the owner of euthanasia greater than the harm to the animal?” [23] (p 73), whereas Herfen et al. [42] provide an algorithm that includes financial reasons as ultima ratio for euthanasia. The latter also go along with Yeates’s [23] suggestion of a contextually justified euthanasia as the algorithm takes the actual living circumstances of the patients and owners into account—and thereby spares the veterinarian the feeling of having performed euthanasia for morally wrong reasons. Furthermore, most of the legal provisions present a great scope for interpretation as they lack precise definitions of criteria for decisions (“good welfare”, “prolonged death”, “compromised welfare”, “continuing to live would be worse than death”, “the animal’s best interest”, etc.) in many cases. On the one hand, they present a necessary legal framework for professionals; on the other hand, they shift the responsibility to both veterinarians and companion animal owners to make the (ethically) right decision in each individual case.

## 4. Intermediate Result

Before discussing the matter, a brief classification of the various uses and definitions of euthanasia is provided (see Figure 1).

The more criteria are added to the definition, the less cases classify as euthanasia. The classification starts from the very rough and unspecific definitions that only focus on the *method* of killing animals. Euthanasia then simply means killing (a) executed by a veterinarian and (b) with a method that avoids pain and distress; in these definitions, the reasons or the justification of the action are not considered at all.

Most definitions of euthanasia include further reflection of the motives of humans involved. They then specify euthanasia by contrasting it to other types of killing animals by mentioning the special interests behind the action. “Proper Euthanasia” [24] (p 21) is considered to be in the animal’s interest, whereas other occasions of killing animals (hunting or slaughter) are purely based on human interests. There is a great variety of theories of what may be considered to count as an interest of an animal, e.g., weighing death against suffering, and judging it to be the minor evil.

Definitions within the context of theories that deny animals own interests to carry on living (for whatever reason), may only refer to the painless character of the action in itself.

An even broader type of definition may also include the interest of the animal’s owner. In case the ending of an animal’s life is not beneficial for the animal itself, these definitions would also take into account the perspective of the owner. The owner may wish to keep his or her animal alive but is unable to provide the necessary resources (contextually justified) or he or she simply wants to end the animal’s life and his or her desires are simply considered to be overriding: Euthanasia then would be called “convenient”.

Additionally, there are cases of public interest, such as the elimination of potentially dangerous animals, the killing of surplus animals in laboratories and shelters or the financial burden of livestock-attacking dogs. There is a broad spectrum of views on many animal-related issues, such as animal research and testing, breeding of certain breeds, killing animals in shelters, etc. However, we refer to procedures that are performed for financial reasons or safety reasons and thereby supposedly in the interest of the public. Here, neither the owner nor the companion animal has an interest in ending the animal’s life, but societal aspects justify what is still subsumed under “euthanasia”. For those reasons, it can be helpful to establish a new terminology, including cases of morally justifiable killing which are not cases of euthanasia [20].

## 5. Discussion

In light of the ethical and legal aspects, it is possible to systematically look at the sources of challenges in EOL decisions in small animal practice.

### 5.1. It Is Possible that Practical Constraints Are Guiding EOL Decisions, While Veterinarians Feel They Should Not

There are reasons why Yeates [23] suggests a concept of “contextually justified” euthanasia and why Herfen et al. [42] develop an algorithm that includes the ultima ratio justification for financial reasons. The conflict between moral arguments favoring the animal’s life to continue and circumstances like time, effort and money that finally lead to euthanasia in spite of what is considered to be in the animal’s best interest is frequently mentioned when discussing moral stress among veterinarians (e.g., [1,9]). If this gap between personal ethical values and practical constraints was the main reason for the numerous cases of burnout and suicide among veterinarians, there should be a significant difference between those countries in which euthanasia is more strictly regulated and those in which convenience or crowded shelters are legitimate reasons for killing animals. Although precise figures are missing for many countries, there are, however, clear hints that cases of burnout and suicide are linked to the veterinarian profession worldwide [2,3,9,48,49]. While the abovementioned tools can provide some relief to those that can acknowledge the argumentative power of certain external circumstances, there is still the option not to euthanize if feeling uncomfortable about an owner’s request [12]. This, again, presents a field of tension for a veterinarian who defines himself as the animal’s advocate but at the same time as a service-provider to a client.

### 5.2. It Is Possible that Veterinarians Have a Fundamental Problem with Killing

Grounded in the quasi-universal human intuition that killing is morally wrong, veterinarians might feel some discomfort when putting an end to a companion animal’s life. In an Austrian study, veterinarians agreed to the statement “I see euthanasia as an unavoidable evil in my responsibility”. Furthermore, they were ambivalent regarding the statement “I am still not used to euthanizing animals” [9] (p 5). As parties being responsible for the decision, both owners and veterinarians report feeling guilty after euthanizing a pet, even if they are convinced it was an overall good decision [50,51]. This feeling is confrontedwith euthanasia being a common service in veterinary practice and therefore a procedure that is taught in veterinary schools and expected of a professional veterinarian. Furthermore, euthanasia is also legally defined and thereby supported as something acceptable. The commonly perceived intertwinement of legal norms and moral evaluation (see, e.g., Feldman [52] ) potentially confronts veterinarians with the impression of simultaneously doing something right (according to law) and wrong (according to their moral intuition). The lack of awareness for different accounts of animal death and welfare and especially a lacking access to what is ubiquitously referred to as “the animal’s best interest” might present a difficulty for veterinarians to explain their discomfort precisely or to dismiss it with a clear positioning within the possible spectrum of accounts.

### 5.3. It Is Possible that Veterinarians Think Euthanasia Is Acceptable, but They Are Not Able to Pinpoint Justifications

Intuitively, that is what might be expected of a veterinarian by a companion animal owner: The veterinarian gives a clear diagnosis and prognosis and some advice on how to come to a good EOL decision. If a veterinarian feels confident to decide intuitively in EOL situations and without precise criteria, the decision might still be acceptable for her and for the patient’s owner. If, however, she decides on the basis of what is legally required, she can be torn between contradictory and unclear statements: Is this euthanasia morally acceptable because the animal is free of pain and discomfort? Is the decision “appropriate for the age” [43] (p 3), and why is age an important factor? Data suggest that veterinarians indeed consider factors, like “Did the animal have a qualitatively and quantitatively positive life?” and “How is the animal’s future perspective?” [53] (p 210) [translated from German by the first author], but they do not further specify how they proceed in his weighing process or which factors give weight in favor of one decision or another. If the best explication for a decision is pointing to a law or guideline that is not clear regarding its normative assumptions or even contradictory, there is a clear challenge for veterinarians to create transparency about their criteria.

### 5.4. It Is Possible that Veterinarians Feel Mentally Torn between the Patient’s, the Owner’s, and the Public’s Interests

Not only the owners who are unwilling or unable to pay but also those who do not want to let go of their companion are part of small animal practice. Apart from that—depending on a country’s circumstances—crowded shelters, biting and game-killing dogs, stray animals, injured free-living animals, or the owner’s convenience are also part of veterinarian practice. Cases of aggressive fighting dogs injuring people, for example, often receive huge media attention (see, for example, [54]). A veterinarian who is acting for reasons of public safety in those cases also euthanizes a dog against the dog’s presumed interest.

At the same time, progress in veterinary medicine holds the potential for overtreating patients [55], an issue that has become an important concern in veterinary practice [56]. Owners who consider their companion animals as family members might want to make use of every possible treatment to keep their pet alive as long as possible—and presumably against the animal’s own interest in cases of an unfavorable prognosis. Again, tools might be helpful for veterinarians for weighing the interests of different parties.

## 6. Conclusions

The plurality of aspects coining the ethical discourse of animal death is not fully reflected in the veterinarian world—neither in laws and guidelines nor in small animal practice. Knesl et al. [10] state that “Euthanasia is an emotional, psychological, and economic issue that every veterinarian must wrestle with” (p 5). While this, unfortunately, seems to be true, it must be emphasized that euthanasia is also an ethical issue that veterinarians must deal with. They are bound to laws and advised by guidelines that are, although not explicitly, mainly based on pathocentric assumptions, i.e., the veterinarians’ attention is directed to the animals’ suffering. In a society that increasingly considers companion animals as family members, though, EOL situation and the value of their lives as such are gaining attention. Besides those individual considerations, research on and specializing in severe diseases in animals might be neglected as long as euthanasia presents an “easy way out”. This context is additionally supported by the American Animal Hospice Movement that provides a spectrum of services around EOL care but also alternative options to euthanasia. (For an elaborate discussion on animal hospice, see Joswig [17].) Finally, euthanasia does present a unique option for veterinarians to bring relief to both the patient and the owner in many cases and leaves the veterinarian with a powerful tool that physicians in human medicine are missing.

The increasing multilayer challenges regarding EOL decisions in small animal practice ask for the following:Psychological support of veterinarians (and pet owners) in EOL decisions.An adapted “toolbox” containing, for example, decision trees or handouts on a deliberate framework [10]. These tools should be implemented in teaching in veterinary schools, and their use should be demonstrated by experienced veterinarians (integrated ethics teaching).A basic training in philosophical assumptions underlying different accounts on animal death, suffering, and welfare. If the veterinarians are familiar with the theory, their own positioning might be facilitated.A more differentiated training in communication, especially in EOL situations.An update in animal law regarding underlying assumptions about animal life, death, suffering and welfare.

At the same time, more practice-oriented work on the meaning of death for nonhuman animals and platforms for the intersection of veterinary ethics and practice could be beneficial for those veterinarians who experience their professional task as burdensome and those animal ethicists dealing with the challenges only theoretically.

## Figures and Tables

**Figure 1 animals-10-00124-f001:**
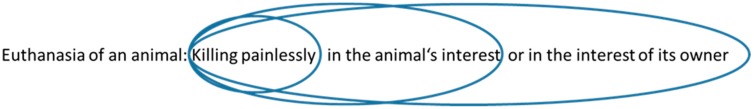
Definitions of euthanasia.

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
