# Peer review of "Philosophy of a “Good Death” in Small Animals and Consequences for Euthanasia in Animal Law and Veterinary Practice"

_animals, 2020, doi:10.3390/ani10010124_

Round 1

Reviewer 1 Report

The paper clearly presents an accurate analysis of the consequences for euthanasia in animal law and veterinary practice. The paper is appreciable for the important reflections that are stimulated in the reader.
For what concerns veterinarians, there is currently a "moral" growth of this category with substantial differences between veterinarians dealing with animal welfare and behavior and generalist veterinarians.
Differences are also present in the training of veterinary students (see Mariti et al., Familiarity in working with livestock decreases the odds of positive attitudes towards non-human animals and their welfare among veterinary students in Italy. Animals 2018; 8 ( 9), 150; Menor-Campos et al., Attitudes to Animals of Three European Veterinary Medicine Schools in Italy and Spain. Anthrozoos. 2019; 32 (3), pp. 375-385; ).

Finally I suggest replacing the word "vivisection" with "scientific experimentation" to avoid the negative connotations that the term vivisection has rightly acquired over the years.

Author Response

Dear Reviewer,

thank you for your valuable remarks. We added some of the literature you suggested concerning veterinary education and replaced “vivisection” with “research experiments”.

Best regards

Kirsten Persson

Reviewer 2 Report

Thank you for the opportunity to review this paper. I am not a veterinarian or a philosopher, but have strong interests in practical ethics, ethical review and refinement (including humane endpoints and humane killing) with respect to lab animals. All the comments below are made in that context.

General comments

Overall, the English is very good, but the draft could do with a critical read by another English speaker as there are some grammatical slips. Examples: line 31 should be ‘reasonS’; line 40 ‘reasons of FOR’, lines 55-58 should say ‘causeS’ and the final ‘are focused on’ does not make sense.

From the perspective of someone involved in laboratory animal welfare, within animal research and testing it is common (although not universal) for a distinction to be made between ‘euthanasia’ and ‘humane killing’.  In these cases, euthanasia is used to describe an act of killing an animal because a humane endpoint has been approached, or reached, and the animal is suffering at levels beyond those permitted in the project licence – a ‘mercy killing’. It would also be euthanasia if the animal was suffering from a veterinary condition. Where an animal is killed as part of a procedure, or because they are surplus, this is referred to as humane killing. Although the animal’s experience of the killing process is the same, many people feel that the term euthanasia is a euphemism if the killing was not for the animal’s benefit. (Some scientists also use the terms ‘cull’ or ‘sacrifice’ in applications, which is usually picked up by ethics committees!) You mention intention in line 205, and this comes to the fore in the laboratory.

I have gone into detail about this because you touch on animal research and testing in several places, e.g. lines 116, 142 (within the encyclopaedia definition – as an aside, it is frustrating to see they suggest animals can ‘donate’ tissues, although I know these are not your words), 388. You also mention the distinctions that can be made in a laboratory setting in the footnote below line 393 and say this cannot be discussed in full. I understand that you will not want to add to the length of the paper, but you risk confusing people if you do not establish a clear difference between laboratory and companion animals and the laws that regulate their treatment. 

For example, in lines 387 to 391 you need to delete the obsolete EC reference (see below) and cite

something relevant from Directive 2020/63/EU (https://eur-lex.europa.eu/legal-content/EN/TXT/HTML/?uri=CELEX:32010L0063&from=EN), Article 6 refers to ‘killing’ and Recital 14 says ‘the methods selected should avoid, as far as possible, death as an end-point due to the severe suffering experienced during the period before death. Where possible, it should be substituted by more humane end-points using clinical signs that determine the impending death, thereby allowing the animal to be killed without any further suffering’. Appropriate ‘methods of killing animals’ are listed in Annex IV. The term ‘euthanasia’ does not actually appear in the Directive.

Specific comments

Lines 73-74 – does decreasing empathy apply to both male and female students? There are a few papers on this suggesting it is less of an issue for women, e.g. https://veterinaryrecord.bmj.com/content/146/10/269

Line 116 – ‘vivisection’ is a value-laden term which is not helpful, especially in a publication aimed at vets who routinely prescribe pharmaceuticals and treatments that have been developed and tested using animals. ‘Research and testing’ or just ‘research’ is a more appropriate term.

Lines 217-221 – you need to mention the concepts of a life not worth living, a life worth living, and a good life here. David Mellor has published on this, e.g. you could cite this paper https://www.mdpi.com/2076-2615/6/3/21 and it is easy to find others.

Line 313 – you could consider mentioning that assessing animal welfare, and quality of life, has progressed furthest within animal research, as in this guidance document: https://ec.europa.eu/environment/chemicals/lab_animals/pdf/Endorsed_Severity_Assessment.pdf. See also the chapter on assessing harms in this document: https://assets.publishing.service.gov.uk/government/uploads/system/uploads/attachment_data/file/675002/Review_of_harm_benefit_analysis_in_use_of_animals_18Jan18.pdf. I agree that there is no way of directly assessing quality of life, and it is good to see you say this instead of talking about ‘measuring’ this which is impossible in my view.

Line 316 – presumably owners are also biased by their own emotional links with (and ‘love for’) the animal?

Line 372 – the EU recommendations for euthanasia of experimental animals do not relate to an Animal Welfare Act. Laws that regulate animal research and testing are enabling acts, because they enable licensees to conduct procedures that would otherwise render them liable for prosecution under animal welfare legislation. Also, the 1997 document is not used any more because it relates to a previous Directive. Suggest delete this reference.

Line 391 – the way the second sentence follows the first in the line implies that the guidance on experimental animals includes dangerous dogs, which is confusing (this point will be obsolete if you do the above edits, but it illustrates the importance of distinguishing between the legislations).

Lines 415-416 – I really like this diagram.

Line 419 – please mention that it is important to minimise distress during euthanasia, not just pain.

Line 438: killing surplus animals in laboratories is not in the public interest, because (i) it means money has been wasted in housing and caring for animals who should never have existed and (ii) it is a matter of concern for many people, who believe that animals have intrinsic worth (Directive 2010/63/EU also acknowledges this). The public is also very concerned about animals being killed in shelters and many people do not agree with raising animals for sporting purposes. Killing potentially dangerous animals is also controversial, as in the campaigning against Breed Specific Legislation in the UK. Suggestions for less controversial examples are humanely killing: animals who have harmed humans or other animals such as companion animals or livestock; animals who are diseased, where it has been reliably predicted that killing these individuals will prevent a disease outbreak (e.g. avian influenza); wild animals who are injured or sick, suffering and are judged unlikely to recover.

Line 510 – game ‘livestock-attacking’.

Line 519 – in the UK at least, over-treatment has become a big issue among the veterinary profession. You could consider developing this slightly, or including a couple more references, e.g. https://veterinaryrecord.bmj.com/content/182/23/664.

Line 534 – ‘research on and specialising in severe diseases in animals might be neglected’ – what are your views on this? My initial response is that this may not be a bad thing, as such research would require creating severe animal models, prolonging the lives of severely ill animals, and/or having ‘control’ animals who would be denied treatments. Euthanasia may be a greater good! Can you express and justify your views on this?

Author Response

Dear Reviewer,

Thank you for your insightful remarks, especially on animal research and testing.

As the focus of our paper is on small animal practice, we touched on killing in animal research and testing only for the purpose of demarcation.

We corrected the remaining spelling and grammar mistakes.

To answer your specific comments:

Lines 73-74 – does decreasing empathy apply to both male and female students? There are a few papers on this suggesting it is less of an issue for women, e.g. https://veterinaryrecord.bmj.com/content/146/10/269

We added “some” to indicate that it does not apply to all students.

Line 116 – ‘vivisection’ is a value-laden term which is not helpful, especially in a publication aimed at vets who routinely prescribe pharmaceuticals and treatments that have been developed and tested using animals. ‘Research and testing’ or just ‘research’ is a more appropriate term.

We changed it to “killing in research and testing”

Lines 217-221 – you need to mention the concepts of a life not worth living, a life worth living, and a good life here. David Mellor has published on this, e.g. you could cite this paper https://www.mdpi.com/2076-2615/6/3/21 and it is easy to find others.

We agree that Mellor’s account on animal welfare is valuable, however, it is not directly linked with Cholbi’s life comparative account. In our opinion, the aspect you mention is better placed in chapter 2.3.2 (different accounts on animal welfare), which is why we added Mellor’s paper, there.

Line 313 – you could consider mentioning that assessing animal welfare, and quality of life, has progressed furthest within animal research, as in this guidance document: https://ec.europa.eu/environment/chemicals/lab_animals/pdf/Endorsed_Severity_Assessment.pdf. See also the chapter on assessing harms in this document: https://assets.publishing.service.gov.uk/government/uploads/system/uploads/attachment_data/file/675002/Review_of_harm_benefit_analysis_in_use_of_animals_18Jan18.pdf. I agree that there is no way of directly assessing quality of life, and it is good to see you say this instead of talking about ‘measuring’ this which is impossible in my view.

We agree that a lot more could be said about animal welfare and quality of life. As our focus is on end-of-life issues in small animal practice, however, we would rather not add QoL approaches in animal research and testing to the paragraph. After all, the “best possible quality of life”, as it reads in the first document you suggest, is still something quite different for a laboratory animal in comparison to a companion animal.

 For the same reasons we avoid an exhaustive presentation of specific QoL accounts in farm animal welfare.

 Line 316 – presumably owners are also biased by their own emotional links with (and ‘love for’) the animal?

We added this aspect in a short sentence. Just as a short remark: In human medical ethics it is often also the right (and burden!) of a patient’s family to make decisions in EOL situations, despite – and because – of their close links.

Line 372 – the EU recommendations for euthanasia of experimental animals do not relate to an Animal Welfare Act. Laws that regulate animal research and testing are enabling acts, because they enable licensees to conduct procedures that would otherwise render them liable for prosecution under animal welfare legislation. Also, the 1997 document is not used any more because it relates to a previous Directive. Suggest delete this reference.

We modified the paragraph and added a sentence referring to the new directive, but we decided to keep the former directive, too, as it shows the unusual inclusion of age as a criterion.

Line 391 – the way the second sentence follows the first in the line implies that the guidance on experimental animals includes dangerous dogs, which is confusing (this point will be obsolete if you do the above edits, but it illustrates the importance of distinguishing between the legislations).

We modified the sentence.

Lines 415-416 – I really like this diagram.

Thank you.

Line 419 – please mention that it is important to minimise distress during euthanasia, not just pain.

Yes, we added that.

Line 438: killing surplus animals in laboratories is not in the public interest, because (i) it means money has been wasted in housing and caring for animals who should never have existed and (ii) it is a matter of concern for many people, who believe that animals have intrinsic worth (Directive 2010/63/EU also acknowledges this). The public is also very concerned about animals being killed in shelters and many people do not agree with raising animals for sporting purposes. Killing potentially dangerous animals is also controversial, as in the campaigning against Breed Specific Legislation in the UK. Suggestions for less controversial examples are humanely killing: animals who have harmed humans or other animals such as companion animals or livestock; animals who are diseased, where it has been reliably predicted that killing these individuals will prevent a disease outbreak (e.g. avian influenza); wild animals who are injured or sick, suffering and are judged unlikely to recover.

We have a different understanding of “public interest” here. In our understanding, surplus animals in laboratories are part of the basic concept of animal research and testing (if they could be avoided, they should be, of course). If animal research is legally and publicly accepted as such, surplus animals are an unavoidable aspect. While it is true, though, that there is public disagreement on animal research in general and also on the necessity of specific animal experiments, there is a financial interest not to keep surplus animals alive until they die (which would be a waste of money) but to kill them if they are no longer needed. We added a footnote to differentiate between the diverse interests of the public and general public interests like safety and financial aspects.

Line 510 – game ‘livestock-attacking’.

Changed that.

Line 519 – in the UK at least, over-treatment has become a big issue among the veterinary profession. You could consider developing this slightly, or including a couple more references, e.g. https://veterinaryrecord.bmj.com/content/182/23/664.

Thank you for the reference. We added it.

Line 534 – ‘research on and specialising in severe diseases in animals might be neglected’ – what are your views on this? My initial response is that this may not be a bad thing, as such research would require creating severe animal models, prolonging the lives of severely ill animals, and/or having ‘control’ animals who would be denied treatments. Euthanasia may be a greater good! Can you express and justify your views on this?

It depends. My answer is very general as I am not a specialist on animal research: After all, we do research on sever animal models for human purposes, too, which does not (necessarily) result in a benefit for any nonhuman animal. However, it would be possible to limit research to those animals who are already severely ill. Having a control group with the standard treatment (which could be palliative) and one with a new potential treatment could result in new medications or other treatment options for severely ill patients and in an alternative to euthanasia. The aspect of overtreatment always needs to be reflected, of course, and it is certainly a weighing process.

The fact that we think it might be better to choose euthanasia than to do research on severe illnesses already tells us something about our account of the meaning of death (compared to suffering) for animals, in my opinion.

Finally, this remark was just an aspect to start a discussion/reflection rather than stating a fact.

Than you for your review and best regards

Kirsten Persson